# Hoof Expansion, Deformation, and Surface Strains Vary with Horseshoe Nail Positions

**DOI:** 10.3390/ani13111872

**Published:** 2023-06-04

**Authors:** Vanessa E. Dahl, Ellen R. Singer, Tanya C. Garcia, David A. Hawkins, Susan M. Stover

**Affiliations:** 1Department of Animal Science, University of California, Davis, CA 95616, USA; 2Sussex Equine Hospital, Ashington, West Sussex RH20 3BB, UK; singer844@btinternet.com; 3Department of Surgical & Radiological Sciences, School of Veterinary Medicine, University of California, Davis, CA 95616, USA; tcgarcia@ucdavis.edu (T.C.G.); smstover@ucdavis.edu (S.M.S.); 4Department of Neurobiology, Physiology and Behavior, University of California, Davis, CA 95616, USA

**Keywords:** horseshoe, hoof, equine, biomechanics

## Abstract

**Simple Summary:**

Horseshoes are placed on the hooves of horses to prevent excessive wear on the hoof when horses are ridden or used for work. Typically, horseshoes are attached to the bottom of the hoof with nails. Although nails have been used for centuries, we sought to understand whether nailing a shoe to the hoof would adversely affect hoof expansion that normally occurs when weight is borne on the limb. Horseshoes were attached to cadaveric hooves by nails in three different sets of positions. Weight bearing was simulated while hoof expansion and compression and tension on the surface of the hoof were measured. We found that placing nails closer to the back of the hoof limited hoof expansion and caused changes in compressive and tensile forces on the surface of the hoof. Limiting hoof expansion interferes with the hoof’s ability to absorb concussion. Over time, the observed changes in compression and tension may cause abnormal changes in hoof shape that can affect soundness and performance.

**Abstract:**

Racehorses are susceptible to underrun heel hoof conformation. Racehorses are often shod with nails placed toward the heel. It is unknown if palmar nails restrict or alter hoof deformation in a manner that could promote the development of underrun heel conformation over time with repeated loading. To determine how the addition of palmar nails affects heel deformation during limb loading, hoof expansion and hoof wall deformations were quantified using rosette strain gauges and kinematic markers during in the vitro limb loading of cadaveric limbs that simulated midstance for walk, trot, and canter loads. Nail treatments used to attach a horseshoe to the hoof included: toe nails (T), toe and quarter nails (TQ), and toe, quarter, and heel nails (TQH). The effects of nail treatment on heel expansion and hoof wall deformations were assessed using repeated measures analysis of variance (*p* < 0.05). Nails placed palmar to the quarters of the hoof decreased heel expansion (*p* < 0.001). Heel nails resulted in the largest changes in hoof wall principal strain directions distally. The application of nails palmar to the hoof quarters alters hoof wall deformation during limb loading. The continued loading of the hoof with palmer nails could alter hoof conformation over time.

## 1. Introduction

Heel expansion and elastic structures within the foot provide some mechanisms for attenuating ground impact forces along with bones and joints in the distal part of the limb [1,2]. Horseshoes are used to protect the hoof from excessive wear. However, attaching a rigid shoe to the hoof could impair normal hoof mechanics by limiting hoof expansion during interaction with surfaces [3]. The chronic impairment of hoof expansion and the alteration of hoof wall load distribution could affect hoof growth and conformation [4]. Altered deformations applied chronically and repetitively during racehorse training provide a potential mechanism for conformational changes that may lead to abnormal hoof conformations.

Abnormal hoof conformations have been associated with injuries and lameness. One type of abnormal hoof conformation is the underrun heel, defined as a heel angle at least 5° lower than the toe angle [5,6]. A study on Thoroughbred horses showed that the odds of carpal effusion increase as the heel becomes more underrun [7]. An increased risk of catastrophic injury to the suspensory apparatus in racehorses has been associated with underrun heel conformation [8]. Underrun heels may predispose to foot lameness in other sport horses, and excessively low heel conformation may predispose to palmar heel pain [9,10]. If abnormal hoof conformation contributes to injury development, it is important to identify and understand factors that contribute to abnormal hoof conformation and thus increase risk for injuries. 

Hoof wall expansions are influenced by the application of a shoe and by the sites of attachment of the shoe to the hoof wall. The placement of shoes decreases normal hoof expansion when compared to an unshod foot [11]. In a study of shoes removed from racehorses, grooves worn in the shoe by hoof wall motion were shorter and thinner when nails were placed further palmar in the hoof, indicating that the heels were more restricted [12]. The effects of the location of nail placement within a shoe on hoof wall strains during limb loading and the long-term effect on hoof conformation changes are unknown.

The study objectives were to determine the effects of nail placement on hoof wall surface strains, deformation, and expansion during the loading of cadaveric limbs in vitro and to determine if changes are consistent with the development of the underrun heel conformation. We hypothesize that greater constraint of the hoof by a horseshoe attached with nails placed progressively more palmar would increase hoof wall strains, change principal strain directions, and alter hoof deformations in a manner consistent with a mechanism that may lead to the development of underrun heel conformation.

## 2. Materials and Methods

### 2.1. Study Design

The effects of nail placement for horseshoe attachment on heel and quarter expansions, hoof wall (global) deformations, hoof wall (local) surface strains, and fetlock extension were assessed during the loading of cadaveric equine limbs in vitro to simulate the middle of the stance for walk, trot, and canter loads. Treatments included no shoe (NS) and horseshoe with toe nails (T), toe plus quarter nails (TQ), or toe plus quarter plus heel nails (TQH). The order of the three nail treatments was different among individual limbs so that the variance associated with the effect of the order of treatment (e.g., potential stress relaxation of the cadaveric limb during repeated testing) could be partitioned out from the variance associated with the effect of nail placement to minimize the effect of the test order on statistical nail treatment outcomes. In addition, NS treatments were included before and after the assigned nail treatments for comparison with the nail treatments and to determine if repeated limb loading affected outcome measures (ideally, the outcome measures for the first and last NS treatments would not be statistically different). Treatment effects were assessed using a repeated measures analysis of variance (ANOVA).

### 2.2. Limb Specimens

A sample of nine unilateral cadaver forelimbs from nine horses (mean age 14 years ± 4.1 (SD); 5 left, 4 right; 3 female, 3 male, 3 unknown; 3 Quarter Horse, 1 Paint, 1 Andalusian, 2 Thoroughbred, 2 unknown breed) were removed by transection at the level of the middle of the radius to maintain the passive components of the fetlock stay apparatus [9]. The limbs had no evidence of orthopedic disease, history of lameness, or hoof abnormalities. All hooves were of a similar size to accommodate a size 7 Queen XT (Thoro’bred, Inc.; Anaheim, CA, USA) horseshoe. Limbs were stored frozen in water-tight bags at −20 °C until thawed to room temperature before instrumentation and testing.

### 2.3. Limb Preparation

The hooves were trimmed and shod by one Certified Journeyman Farrier to ensure medial and lateral balance was achieved. The shoe was attached to the hoof by the toe, quarter, and heel nails (Figure 1A). Then, the shoe and nails were removed, and the limb and hoof were instrumented with kinematic markers and strain gauges. A silicone-based hoof packing material (Sil-Pak, Vettec Hoof Care, Pomona, CA, USA, Stiffness: 2 × 10^6^ N/m) filled the solar surface of the hoof to the level of the weight bearing surface of the unshod or shod hoof. Nails were placed in the pre-existing holes for respective treatments. Due to the location of the hoof markers and strain gauges, the need to maintain the shoe in the same position on the hoof for all treatments, and the likelihood that driving nails in an instrumented hoof would detach the strain gauges from the hoof, the same nail holes were used from when the shoe was initially fitted for all nail treatments. Since the nails were removed and replaced according to the treatment order assignment for each limb, the nails were not clenched.

### 2.4. Limb Instrumentation

The fetlock angle was quantified using kinematic markers on mediolateral transfixation pins located through the centers of the proximal and distal aspects of the third metacarpal bone (MC3) and proximal phalanx (P1) (Figure 1D). Dorso-palmar radiographs of the instrumented limb were made for the conversion of the marker positions to the MC3 and P1 longitudinal axes (58 kVp, 10.2 mAs, 1 m film-focal distance, Digital processor: Mark III, Sound-Eklin DR, Carlsbad, CA, USA, Generator: HF100/30+, MinXray, Inc., Northbrook, IL, USA).

Global hoof wall deformations were measured with kinematic markers applied to the medial and lateral aspects of the heels and quarters, at approximately 33% and 66% of the distance from the hairline to the solar margin of the hoof (Figure 1B,C). Spherical balls (3/8” diameter polytetrafluoroethylene balls (McMaster-Carr, Atlanta GA)) covered by silver reflective tape (Scotchlite 3M, St. Paul, MN, USA) were attached to 4.8 mm diameter Steinmann pins embedded in the stratum medium of the hoof wall. 

Local hoof wall deformations were measured (rectangular, 45° three-element, rosette strain gauges, standard Wheatstone bridge configuration with temperature compensation, C2A-06-031WW-350, VPG, Raleigh, NC, USA) on the lateral side of the hoof wall at six locations: three proximal-distal locations (roughly 25%, 50%, and 75% of the total length, as measured from the hairline to the solar margin of the hoof along the direction of horn tubules) on the quarter (widest part of the hoof) and heel (halfway between the quarter and palmar aspect of the heel). Locations are defined as the proximal heel (PH), middle heel (MH), distal heel (DH), proximal quarter (PQ), middle quarter (MQ), and distal quarter (DQ) (Figure 1B). Gauges were attached using cyanoacrylate (M-bond 200, VPG, Malvern, PA) after the hoof wall was abraded (360 grit sandpaper), degreased (acetone), and chemically cleaned (M-Prep Conditioner A and M-Prep Neutralizer 5A, VPG, Malvern, PA, USA).

### 2.5. Mechanical Testing

Mechanical loading was performed in a servohydraulic material testing system (Model 662.10A-08, MTS Systems Corporation, Eden Prairie, MN, USA, Model 809; MTS Systems Corp., Minneapolis, MN, USA) equipped with an axial-torsional load transducer (axial load range 220 kN, resolution 22 N, torsional range 2.5 kNm, resolution 0.25 Nm) and software (Multipurpose Testware, TestStarII v4.0c, MTS, Eden Prairie, MN, USA). The proximal end of the radius was potted in an aluminum cylinder with polymethylmethacrylate (PMMA, Co Tray Plastics, GC America Inc., Alsip, IL, USA), which was secured to the mechanical testing system. The foot was secured to the system actuator via a translation table on linear bearings to allow the hoof to translate dorsally during limb loading, which maintained the radius and metacarpal bones parallel to the axis of loading. The initial positioning was standardized by obtaining a physiologic (mean 210°) palmar fetlock angle at a load of ~700 N. The limb was preconditioned by 200 sinusoidal cycles (700–1800 N) at 0.25 Hz under axial displacement control with rotation displacement fixed. Subsequently, the limb was similarly loaded (700–6700 N) for three sinusoidal cycles at 0.25 Hz (~3350 N/s) to capture the peak vertical forces reported for stance (1800 N), walk (3600 N), trot (4600 N), and canter (6700 N) [13,14], while axial load and displacement data were captured at 60 Hz. Strain data were collected at 20 Hz to allow for the multiplexing of the 24 channels. Video images (Photron Fastcam PCI cameras and Fastcam Viewer Photron, San Diego, CA, USA) were acquired at 60 Hz from palmaromedial and palmarolateral positions (S-PRI, AOS Technologies, AG, Baden Daettwi, Switzerland) to capture the positions of kinematic markers in a calibrated field. 

### 2.6. Data Reduction

Data were analyzed for 3600, 4600, and 6200 N loads. Since one limb did not achieve 6700 N load, 6200 N was used as the experimental load for all limbs. Principal strain magnitudes and directions were calculated based on equations by Schajer [15]. The hoof wall deformations, hoof expansions, and fetlock angle (palmar angle between MC3 and P1) were determined from the positions of the markers in video images (Vicon Motus 10, Contemplas GmbH, Kempten, Germany). The locations of bone markers were transformed to bone-fixed virtual markers along the MC3 and P1 longitudinal axes. Six chord lengths on the lateral side of the hoof wall (proximal (Pr), distal (Di), dorsal (Do), palmar (Pa), proximodorsal to distopalmar (PrDo-DiPa), and proximopalmar to distodorsal (PrPa-DiDo)) were measured (Figure 1C). Quarter and heel expansions were calculated by the change in chord distances between the respective medial and lateral markers (Figure 1D). Differences from the stance load (1800 N) to 3600 N, 4600 N, and 6200 N loads are reported. A forecast technique (Excel, Microsoft, Redmond, WA) was used to interpolate the strain and kinematic data for the desired loads. The linear and angular resolutions of kinematic markers were 0.01 mm and 0.11°, respectively.

### 2.7. Statistical Analysis

An analysis of variance (Proc Mixed, SAS 9.4) that accounted for repeated measures was used to assess the effects of being barefoot (NS) and shod with shoes of different nail positions (T, TQ, TQH) on the hoof wall strains, deformations, heel expansion, and fetlock angle using a commercial statistical program (SAS, SAS Institute, Cary, NC, USA). Differences between the initial and final NS treatment were not detected, so the initial NS measurements were used for the NS data points. The treatment, load, order, and treatment*load interaction were treated as fixed effects. The horse was treated as a random effect. The treatment*load interaction was not statistically significant (*p* > 0.05) for any variable and was therefore removed from the ANOVA model. To determine whether the order influenced the outcomes, ANOVA was performed on the data without NS treatment. If the order was found to be significant, and the residuals were normally distributed (Shapiro–Wilks test), the model assumptions were assumed to be valid. If the order was not found to be significant, the order effect was removed from the model, the initial NS treatment was added back in, and the model was re-run. 

The residuals from the analysis of variance (ANOVA) were examined for normality using a Shapiro–Wilks test (Proc Univariate, SAS). The residuals were considered normally distributed if W ≥ 0.9 and the residuals were aligned on the QQ plot. For variables that had ANOVA residuals that were not normally distributed, a box plot (Proc Boxplot, SAS) was used to determine outliers (observations that fell below Q1 − 1.5 IQR or above Q3 + 1.5 IQR). The ANOVA was performed after outlier removal, and the residuals were assessed for normality. For variables with non-normally distributed ANOVA residuals, the Box Cox procedure was used to determine a transformation appropriate for each variable (Proc Glmselect and Proc Transreg, SAS). For variables that had any negative values, a constant (equal to the smallest value plus the absolute value of the difference between the smallest and second-smallest values) was added to all values for the application of the Box Cox procedure. The transformed data were again run through the ANOVA model, and the residuals were assessed for normality. For variables that were still non-normally distributed, raw data were transformed using the Rank procedure (Proc Rank), and ranked data were analyzed using the analysis of variance. A *p*-value < 0.05 was considered statistically significant. Because there were no significant load*treatment interactions, the least squared means and standard error averaged over all loads for the treatment and over all treatments for the load are reported for all variables.

## 3. Results

### 3.1. Fetlock Extension

Fetlock extension increased significantly with an increasing load (Table 1) but was not significantly different among treatments.

### 3.2. Hoof Expansions

The largest differences in expansion were observed in the heel location. Proximal and distal heel expansions increased with increasing loads (Table 1). Expansion approximately doubled from walk to trot and from trot to canter. Lesser magnitudes of expansion were observed in the quarter locations with the proximal expansion greater than the distal expansion. 

The distances between proximal and distal heel markers decreased with the addition of more palmar nails (Table 2), indicating a decrease in expansion. The hoof expansions in the quarter region were smaller compared to the expansions in the heel region and less consistent in the pattern with treatments. 

### 3.3. Hoof Wall Deformations

The Proximal, Distal, and PrPa-DiDo segment lengths, parallel and oblique to the solar aspect of the hoof, respectively, increased with increasing loads (Table 1). The Dorsal, Palmar, and PrDo-DiPa segment lengths, perpendicular and oblique to the weight-bearing surface, respectively, were not different among loads.

Nail treatment differences were observed for the Distal and Palmar segments (Table 2). The Distal and Palmar segment distances were smaller for the TQH treatment than for the NS treatment. 

### 3.4. Hoof Strains

#### Principal Strains

The principal tensile strains decreased for the PH, MH, and PQ gauges, while they increased for the DH, MQ, and DQ gauges as the load was increased; however, only the heel gauges were found to have significant differences between loads (Table 3). The principle compressive and shear strains increased for all gauges with an increased load. All but the DQ gauge had differences between loads for principal compressive strains. During loading, maximum shear strains were greater than compressive strains, which were greater than tensile strains for all but the distal gauges. The distal heel and quarter gauges had tensile strains that were greater than compressive strains but less than maximum shear strains. 

Tensile principal strains for distal heel (DH) and all quarter gauges differed among nail treatments (Table 4). Tensile strains were larger overall for DH and DQ gauges, with the TQH treatment having the largest strains. 

The PH, MQ, and DQ compressive principal strains differed among nail treatments (Table 4). The MQ compressive principal strain most consistently increased with the addition of palmar nails. PH differences varied inconsistently among treatments. 

Maximum shear strains in the MQ and DQ gauges increased with the addition of nails placed more palmar; however, maximum shear strains varied inconsistently among treatments for PH and PQ gauges (Table 4). 

### 3.5. Principal Strain Directions

Principal strain directions did not differ significantly among loads but differed among treatments (Table 5). The principal tensile strain directions for all sites were roughly aligned with the hoof wall tubules for NS through TQ nail treatments (Figure 2A). The greatest differences from the NS treatment were observed for the TQH treatment at the middle gauges and distal quarter gauge locations for principle tensile, compressive, and shear strain directions (Table 6, Figure 2B,C). The increases in the angle of the directions for the principle tensile, compressive, and shear strains were 73°, 57°, and 65°, respectively, from the NS to the TQH treatment. For the lateral aspect of the right hoof, all strain vectors rotated counterclockwise with the addition of more palmar nails. 

Principal compressive strain directions tended to roughly align perpendicular to the hoof wall tubules (Figure 3B). The largest changes in direction occurred with the TQH treatment and were most pronounced in the MH and DQ locations (Table 6, Figure 3B). Compressive strains at the proximal locations had the smallest differences in direction among nail treatments.

## 4. Discussion

Our findings showed that the greater constraint of the hoof by a horseshoe attached with the addition of nails placed progressively more palmar limits heel and quarter expansion, increases hoof wall strain magnitudes, changes principal strain directions, and alters hoof deformation. Generally, tensile principal strains were oriented along the horn tubules, while compressive principal strains were oriented perpendicular to the tubules for the unshod hoof. Hoof wall surface compressive and shear strains increased with limb loading, with the greatest magnitude alteration at the middle heel, distal heel, and middle quarter locations. Additionally, this study included novel locations of strain gauge placement when compared to other studies using strain gauges placed along the hoof wall [16,17].

The hoof and limb behaved in an expected manner under increased loading conditions. The fetlock extension became greater as the load increased on the in vitro limb simulating the middle of the stance, consistent with in vivo loading in live horses [17]. The principal compressive hoof wall strains in this study were larger than the principal tensile strains during loading, consistent with Thomason’s strain experiments in the hoof wall when measured on the lateral aspect of the hoof at the quarters [18,19]. The heel and quarter expansions observed in the current study were consistent with the findings of Douglas [20] as loads increased. The current study demonstrated that the compressive principal strain was greater than the tensile principal strain at the middle heel gauge, which was different than the findings of Bellanzini et al. (2007), who found that gauges placed in the same location recorded a larger tensile than compressive strains; however, this difference could be due to their use of a single foil (uniaxial) gauge, whereas we used rosette gauges [16]. Therefore, Bellanzini’s study was measuring the strain at the site but not providing the principal strain magnitude and direction. 

The finding that the expansion of the heels was greater than the expansion of the quarters may be attributable to three factors. First, greater expansion at the heels results from the normal hoof anatomy in which the hoof wall is the thickest dorsally, gradually thinning and becoming more flexible toward the palmar aspect of the hoof [20,21]. Second, the hoof capsule is an open truncated cone with the opening at the heels, allowing the heels to move further than the quarters. Third, as the distal phalanx moves distally and exerts axial traction on the hoof wall through the laminar junction, the greater contact area of the quarters compared to the heels with the laminae likely contributes to the smaller quarter expansion [16]. 

The current study demonstrated that heel expansion and distal quarter expansion decreased as nails were placed closer to the heel of the hoof, likely interfering with normal hoof behavior. However, a unique finding was that the medial and lateral proximal quarters moved further from each other when nails were applied to the toe and toe/quarter regions of the hoof wall, whereas there were no differences between an unshod hoof and the most constrained condition of TQH. When the distal quarter of the hoof is constrained by horseshoe nails, the proximal aspect of the quarter is forced to expand in response to the load. Hoof expansion is restricted distally by the nails, resulting in the quarters expanding at the proximal aspect, since this is the area that is most able to absorb the diverted force. The cause of the proximal quarter expansion may be related to limb loading and the compression of structures within the hoof capsule exerting a peripherally directed force on the hoof wall near the coronary region [18]. The proximal aspect of the quarter is forced to expand, while the distal aspect of the hoof quarter is constrained by the horseshoe. This expansion of the quarter proximally with the use of nails at the quarter may also be associated with the potential for vertical crack propagation that sometimes occurs in underrun heels [22]. Paradoxically, the addition of heel nails (TQH treatment) was associated with an apparent reversal of proximal quarter expansion changes observed with the T and TQ treatments, appearing more like the unshod condition. This is likely due to a more uniform restriction of the expansion and deformations of the heel and distal quarter regions by the additional heel nails, similar to the uniform lack of restriction the hoof experiences when unshod. 

The current findings of changes in expansion at the proximal quarter location may also be relevant to the distortion of hoof conformation. In our experience, distal curvature of the palmar aspect of the coronary band often accompanies the underrun heel hoof conformation. Mediolateral expansion of the hoof capsule in the region of the proximal portion of the quarters increased with the addition of T and TQ nails; however, it decreased by half with the addition of the heel nails. Conformation of the hoof capsule reflects hoof growth (hoof wall production) at the coronary region. Hoof growth both responds to and reflects hoof loading conditions. In hooves where there is a medio-lateral imbalance, the side where the hoof bears more loads have slower growth compared to the side that experiences less of a load [23]. While hoof conformation is managed at the solar margin through trimming and shoeing, enhancing our understanding of hoof abnormalities requires further investigation into how hoof growth responds to loading and hoof management strategies. It is possible that abnormalities in hoof wall mechanics could contribute to a variety of hoof wall pathologies—for example, quarter cracks and sheared heels.

Several findings associated with the addition of heel nails could contribute to the development of underrun heels. The direction of the principal tensile strain in the distal heel and quarter regions changed the orientation to exert dorsal traction on the distal aspect of the heel when the hoof is fixed to the ground (Figure 3A). Similarly, the orientation of the principal shear strains changed to facilitate shear between the proximal and distal portions of the hoof in a plane perpendicular to the dorsal hoof wall, thus allowing the proximal portion of the heel to move palmarly relative to the distal portion of the heel. Further, the direction of principal compressive strains changes to direct compression on the heel in a distal orientation, potentially forcing the bulb distally. Consistent with this finding is the shortening of the palmar chord on the hoof wall. Additionally, the most extreme limitation of heel expansion occurred at the proximal level of the heel (Figure 3B), which could contribute to additional compression driving the bulb and proximal aspect of the heel distally. These features are consistent with the displacement of the proximal aspect of the heel in palmar and distal directions, a decrease in the heel angle, the narrowing of the heel bulbs, and the distal displacement of the coronary band in the heel region.

Limitations of the current study include the in vitro use of cadaver limbs. Pins placed within the bones and hoof wall precluded the use of live horses and restricted the assessment of deformations to pin locations. The potential effects of pins on the strain results were minimized by placing pins distant to strain gauges, considering Saint-Venant’s principle [24]. Speckle Imaging and Digital Image Correlation (DIC) [25] would be less invasive and provide a more global perspective of hoof deformation, but this technology was not available for this study. DIC provides similar strain results; however, these may be less precise due to the nature of the inconsistency of the application of the speckle pattern. However, the DIC method would show the distribution of strains over the entire speckled surface of the hoof wall and could be useful for further studies. Our hoof wall strain findings were consistent with compressive strain trends observed from other in vivo studies [16], thus supporting the validation of our experimental model. Hoof dimensions including toe and heel angles, hoof wall thickness, and overall size vary among breeds. The breed was not considered in the current study and could potentially influence hoof strains and deformations. The study is limited by the two-dimensional aspects of motion measurements and surface strains. The use of additional markers on the pins would have captured three-dimensional motion and allowed for the determination of both pin rotation and translation, providing more insight into three-dimensional hoof behavior—specifically, whether bulging or depression of the hoof wall contributed to the observed changes in segment lengths. Static measurements were taken with a limb fixed on a metal plate, preventing the measurement of hoof behavior with a natural surface and the evaluation of hoof motions during heel strike and toe off when loaded. However, since the greatest load on the hoof is during the middle of the stance, simulating midstance loads captured information that should be relevant to hoof deformations. Additionally, with the fixed direction of force during the simulation of the mid-stance of gait, variation that occurs with the different ground reaction force patterns associated with each gait is not a factor in the study. A soft pour-in pad was applied to the sole to allow for entire hoof ground contact during loading. The pad material used was the least stiff of the available commercial hoof pad products (2 × 10^6^ N/m) and was less stiff than typical surfaces on which horses are generally worked (dirt approximately 1.65 × 10^5^ N/m, synthetic approximately 2.64 × 10^5^ N/m; unpublished data). The effect of the nail position (T, Q, or H) could not be separated from the effect of the number of nails applied (one, two, or three) because the addition of nails was only performed in the palmar direction. Finally, while there have been no studies, to our knowledge, on the effects of freezing on hoof tissue and its mechanical behavior compared to that of fresh tissue, we acknowledge that there is a potential for freezing and thawing to affect the results. However, since some of our results in strain and hoof expansion measurements do match those that were performed in live horse hooves, we are confident that the effects of freezing on the hoof are minimal.

## 5. Conclusions

In conclusion, shoeing with the addition of nails palmar to the widest portion of the hoof (quarters) reduces hoof expansion and hoof wall deformation at the heels and alters hoof wall principal strain directions in a manner consistent with mechanisms leading to the development of underrun heels and thus injury risk. The practice of placing nails more palmarly in relation to the quarters has been observed in racing horses [12], and injuries in racehorses have been associated with underrun heels [6,7,8]. Placing nails dorsal to the quarters of the hoof may lead to a decrease in underrun heel development. Additionally, the restriction of normal surface hoof deformations may also alter the deeper anatomic structures, contributing to the development of abnormal hoof conformations, foot pain, or related injuries. While nail placement may have a role in the development of an underrun heel, other factors may also contribute to this development, including trimming and shoeing methods, the surface substrate, genetics, and exercise [26]. Additional longitudinal studies using horses that are in work could be performed to test the effects that nail placement has on hoof growth and distortion over time.

## Figures and Tables

**Figure 1 animals-13-01872-f001:**
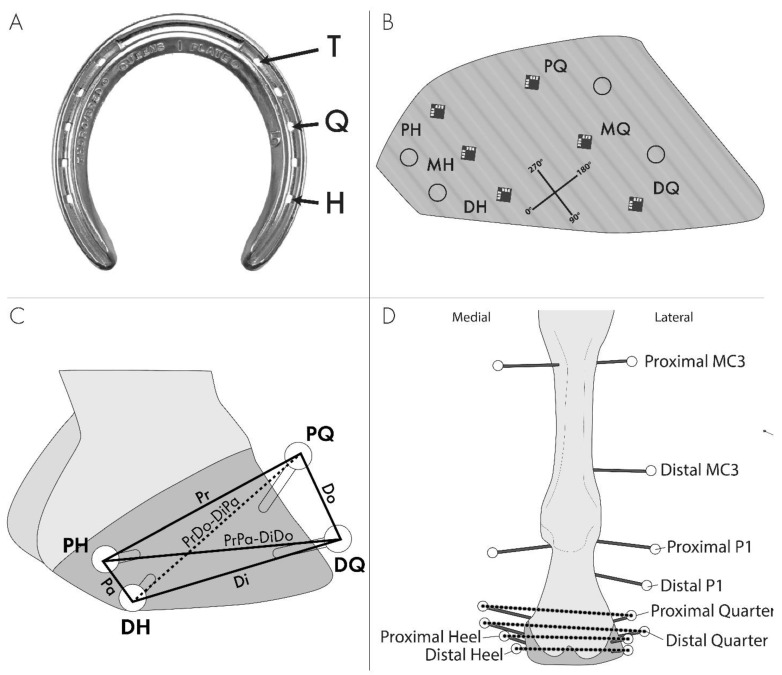
Limb instrumentation. (**A**) Nail treatment locations that were used for a combination of toe (T), toe plus quarter (TQ), and toe plus quarter plus heel (TQH). (**B**) Lateral view of the hoof instrumented with strain gauges, and the coordinate system for strain directions as well as where kinematic hoof markers were placed in relation to the strain gauges. Gauge positions are defined as proximal heel (PH), middle heel (MH), distal heel (DH), proximal quarter (PQ), middle quarter (MQ), and distal quarter (DQ). The 90/270° axis is along the tubules such that the 0/180° axis is perpendicular to the tubules. (**C**) Lateral hoof wall segment definitions for defining lateral wall distortion: Proximal (Pr), distal (Di), dorsal (Do), palmar (Pa), proximodorsal to distopalmar (PrDo-DiPa), and proximopalmar to distodorsal (PrPa-DiDo). (**D**) Schematic illustration of a palmar view of the instrumented limb with transfixation pins and kinematic markers used for measuring the fetlock angle and proximal and distal quarter and heel expansions.

**Figure 2 animals-13-01872-f002:**
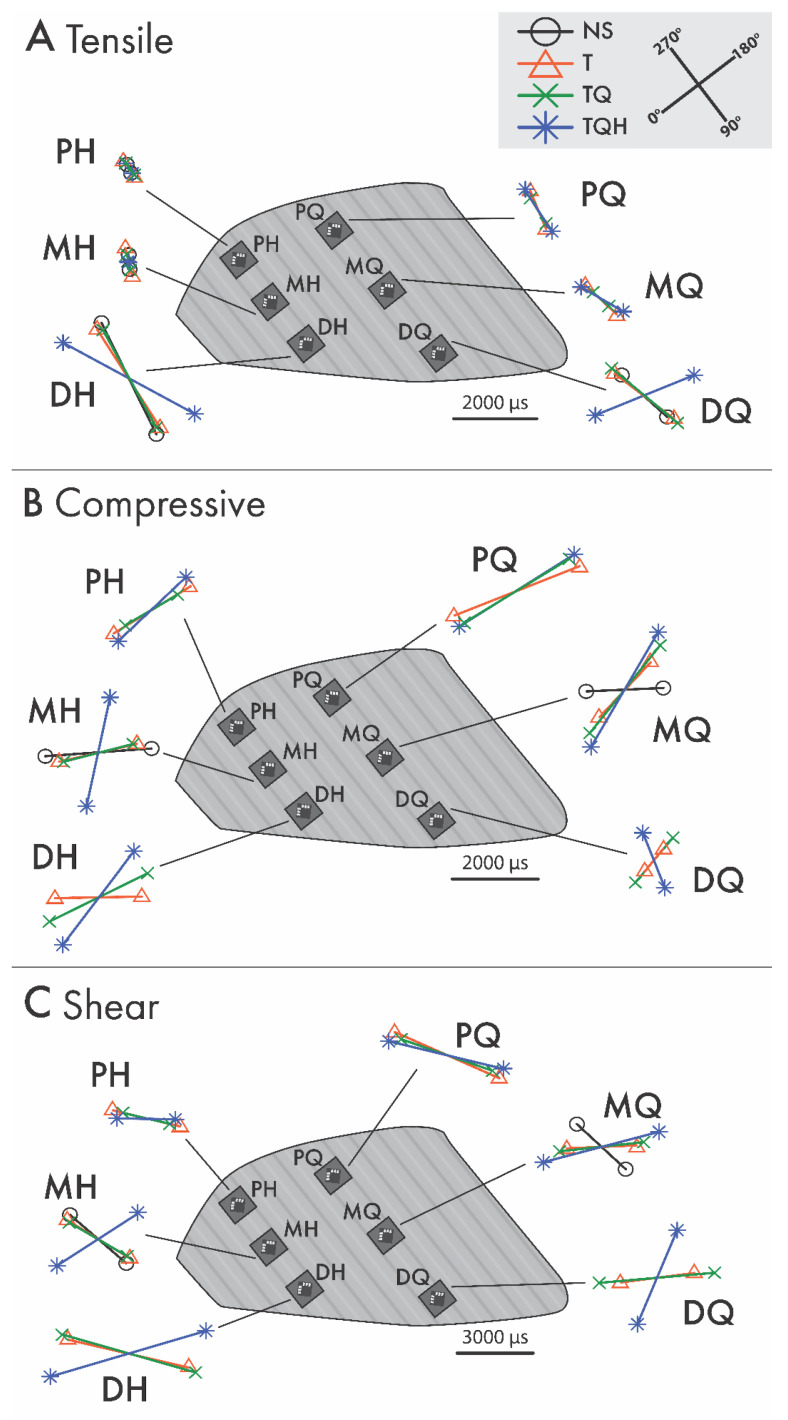
The magnitudes and directions of tensile (**A**), compressive (**B**), and maximum shear (**C**) principal strains are illustrated for each location and treatment on the lateral wall of a right hoof for proximal heel (PH), middle heel (MH), distal heel (DH), proximal quarter (PQ), middle quarter (MQ), and distal quarter (DQ) gauge placement.

**Figure 3 animals-13-01872-f003:**
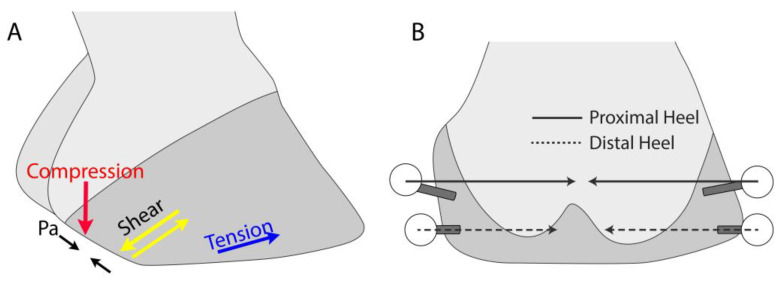
Illustration of the findings associated with the TGH treatment on the lateral (**A**) and palmar (**B**) aspects of the foot that are consistent with the development of the underrun heel. Shortening of the Pa chord, principal tensile strain direction in the DQ gauge location, principal compressive strain direction in the MH gauge location, and principal shear strain direction in the MH and DH gauge locations are consistent with the dropping of the proximal aspect of the heel and the decreasing heel angle. A greater restriction of heel expansion at the proximal aspect of the heels than at the distal aspect of the heels drives the bulbs of the heels under the foot and subjects them to higher compressive forces.

**Table 1 animals-13-01872-t001:** Heel and Hoof Wall Deformation by Load. Fetlock angle, hoof expansion, and wall deformation variables (least square means ± standard error) for 3600 N, 4600 N, and 6200 N loads, when averaged over all treatments. Within a row, values that do not share a superscript are statistically different at *p* < 0.05. *p* values are Type 3 tests of fixed effects for the load in the ANOVA model.

Variable	Load			*p* Value
	3600 N	4600 N	6200 N	
FETLOCK EXTENSION				
Fetlock angle (°) (n = 9)	228 ^a^ ± 0.4	235 ^b^ ± 0.4	244 ^c^ ± 0.4	<0.001
HOOF EXPANSION				
Proximal heel (mm) (n = 9)	0.22 ^a^ ± 0.21	0.53 ^a^ ± 0.21	1.22 ^b^ ± 0.21	<0.0001
Distal heel (mm) (n = 9)	0.35 ^a^ ± 0.08	0.65 ^b^ ± 0.08	1.33 ^c^ ± 0.08	<0.0001
Proximal quarter (mm) (n = 8)	0.25 ^a^ ± 0.08	0.37 ^a,b^ ± 0.08	0.52 ^b^ ± 0.08	0.009
Distal quarter (mm) (n = 8)	0.13 ^a^ ± 0.05	0.21 ^a^ ± 0.05	0.32 ^b^ ± 0.05	0.007
WALL DEFORMATION				
Pr (mm) (n = 7)	0.12 ^a^ ± 0.04	0.17 ^a^ ± 0.04	0.28 ^b^ ± 0.04	0.0001
Di (mm) (n = 9)	0.14 ^a^ ± 0.03	0.19 ^a^ ± 0.03	0.31 ^b^ ± 0.03	<0.0001
Pa (mm) (n = 9)	0.14 ^a^ ± 0.02	0.16 ^a^ ± 0.02	0.16 ^a^ ± 0.02	0.562
Do (mm) (n = 6)	0.70 ^a^ ± 0.57	0.78 ^a^ ± 0.57	0.81 ^a^ ± 0.57	0.052
PrDo-DiPa (mm) (n = 7)	0.47 ^a^ ± 0.38	0.62 ^a^ ± 0.38	0.66 ^a^ ± 0.38	0.168
PrPa-DiDo (mm) (n = 9)	0.18 ^a^ ± 0.06	0.28 ^a^ ± 0.06	0.39 ^b^ ± 0.06	<0.0001

**Table 2 animals-13-01872-t002:** Heel and Hoof Wall Deformations by Treatment. Hoof expansion and wall deformation variables (least square means ± standard errors) for no shoe (NS), toe (T), toe plus quarter (TQ), and toe plus quarter plus heel (TQH) treatments, averaged over all loads. Within a row, values that do not share a superscript are statistically different at *p* < 0.05. *p* values are Type 3 tests of fixed effects for treatment in the ANOVA model. NA = These variables did not have NS included in the model, as the order was found to be significant in the statistical model.

Variable	Treatment				*p* Value
	NS	T	TQ	TQH	
HOOF EXPANSION					
Proximal heel (mm) (n = 9)	1.02 ^a^ ± 0.23	0.91 ^a^ ± 0.23	0.63 ^a^ ± 0.23	0.06 ^b^ ± 0.23	<0.0001
Distal heel (mm) (n = 9)	1.03 ^a^ ± 0.08	0.86 ^b^ ± 0.08	0.71 ^b^ ± 0.08	0.53 ^c^ ± 0.08	<0.0001
Proximal quarter (mm) (n = 8)	0.28 ^a^ ± 0.09	0.45 ^a,b^ ± 0.09	0.55 ^b^ ± 0.09	0.25 ^a^ ± 0.09	0.002
Distal quarter (mm) (n = 8)	0.27 ^a,b^ ± 0.06	0.30 ^b^ ± 0.06	0.16 ^b,c^ ± 0.06	0.16 ^c^ ± 0.06	0.0002
WALL DEFORMATION					
Pr (mm) (n = 7)	NA	0.17 ^a^ ± 0.04	0.21 ^a^ ± 0.04	0.19 ^a^ ± 0.04	0.410
Di (mm) (n = 9)	0.26 ^a^ ± 0.03	0.17 ^a^ ± 0.03	0.21 ^a^ ± 0.03	0.20 ^a^ ± 0.03	<0.0001
Pa (mm) (n = 9)	0.19 ^a^ ± 0.02	0.14 ^a,b^ ± 0.02	0.17 ^a,b^ ± 0.02	0.11 ^b^ ± 0.02	0.041
Do (mm) (n = 6)	NA	0.76 ^a,b^ ± 0.57	1.17 ^a^ ± 0.57	0.36 ^b^ ± 0.57	0.655
PrDo-DiPa (mm) (n = 7)	NA	0.56 ^a,b^ ± 0.38	0.86 ^a^ ± 0.38	0.33 ^b^ ± 0.38	0.148
PrPa-DiDo (mm) (n = 9)	0.34 ^a^ ± 0.06	0.30 ^a^ ± 0.06	0.27 ^a^ ± 0.06	0.24 ^a^ ± 0.06	0.251

**Table 3 animals-13-01872-t003:** Principal Strains Least Squared Means for Load. Least Squared Means microstrain (µε) and standard error for strain magnitudes by load when averaged over all treatments. Within a row, values that do not share a superscript are statistically different at *p* < 0.05. *p* values are Type 3 tests of fixed effects for the load in the ANOVA model. Locations are defined as the proximal heel (PH), middle heel (MH), distal heel (DH), proximal quarter (PQ), middle quarter (MQ), and distal quarter (DQ) (Figure 1B).

Gauge	Load			*p*-Value
	3600 N	4600 N	6200 N	
PRINCIPAL TENSILE STRAIN (µε) lsm ± se				
PH (n = 9)	238 ^a^ ± 53	173 ^a,b^ ± 53	62 ^b^ ± 53	0.0025
MH (n = 9)	328 ^a^ ± 94	243 ^a^ ± 94	−8 ^b^ ± 94	0.0018
DH (n = 9)	1442 ^a^ ± 200	1486 ^a^ ± 200	1592 ^a^ ± 200	0.0295
PQ (n = 8)	510 ^a^ ± 125	497 ^b^ ± 125	427 ^b^ ± 125	0.3263
MQ (n = 9)	453 ^a^ ± 235	466 ^a^ ± 235	471 ^a^ ± 235	0.977
DQ (n = 9)	948 ^a^ ± 125	988 ^a^ ± 125	1020 ^a^ ± 125	0.4352
PRINCIPAL COMPRESSIVE STRAIN (µε) lsm ± se				
PH (n = 9)	−775 ^a^ ± 224	−879 ^a^ ± 224	−1197 ^b^ ± 224	<0.0001
MH (n = 9)	−714 ^a^ ± 251	−971 ^a^ ± 251	−1633 ^b^ ± 251	<0.0001
DH n = 9)	−1045 ^a^ ± 331	−1143 ^a,b^ ± 331	−1476 ^a^ ± 331	0.0132
PQ (n = 8)	−1353 ^a^ ± 414	−1468 ^a,b^ ± 414	−1689 ^b^ ± 414	0.0184
MQ (n = 9)	−994 ^a^ ± 199	−1127 ^b^ ± 199	−1445 ^b^ ± 199	<0.0001
DQ (n = 9)	−557 ^a^ ± 214	−558 ^a^ ± 214	−608 ^a^ ± 214	0.9005
PRINCIPAL SHEAR STRAIN (µε) lsm ± se				
PH (n = 9)	1034 ^a^ ± 233	1103 ^a^ ± 233	1254 ^a^ ± 233	0.0728
MH (n = 9)	1194 ^a^ ± 229	1399 ^a^ ± 299	1864 ^b^ ± 299	0.003
DH (n = 8)	2460 ^a^ ± 386	2612 ^a,b^ ± 386	3073 ^b^ ± 386	0.0576
PQ (n = 9)	2020 ^a^ ± 561	2108 ^a^ ± 561	2198 ^a^ ± 561	0.4815
MQ (n = 9)	1453 ^a^ ± 239	1578 ^a,b^ ± 239	1887 ^b^ ± 239	0.0089
DQ (n = 8)	1787 ^a^ ± 365	1849 ^a^ ± 365	1965 ^a^ ± 365	0.3716

**Table 4 animals-13-01872-t004:** Principal Strains by Treatment. Principal strain magnitudes (least squared means ± standard errors) by nail treatment when averaged over all loads. Within a row, values that do not share a superscript are statistically different at *p* < 0.05. *p* values are Type 3 tests of fixed effects for treatment in the ANOVA model. NA = These variables did not have no shoe (NS) included in the model, as the order was found to be significant upon the initial testing of the model. Locations are defined as the proximal heel (PH), middle heel (MH), distal heel (DH), proximal quarter (PQ), middle quarter (MQ), and distal quarter (DQ) (Figure 1B).

Gauge	Treatment				*p*-Value
	NS	T	TQ	TQH	
PRINCIPAL TENSILE STRAIN (µε)					
PH (n = 9)	115 ^a^ ± 56	233 ^a^ ± 56	147 ^a^ ± 56	136 ^a^ ± 61	0.1673
MH (n = 9)	168 ^a,b^ ± 101	345 ^a^ ± 97	200 ^a,b^ ± 101	37 ^b^ ± 112	0.0674
DH (n = 9)	1489 ^a,b^ ± 203	1395 ^a^ ± 200	1356 ^a^ ± 203	1788 ^b^ ± 220	0.0002
PQ (n = 8)	NA	494 ^a,b^ ± 124	358 ^a^ ± 131	581 ^b^ ± 124	0.0081
MQ (n = 9)	NA	538 ^a^ ± 236	270 ^b^ ± 234	582 ^a^ ± 236	0.0024
DQ (n = 9)	741 ^a^ ± 128	893 ^a,b^ ± 125	1025 ^b,c^ ± 128	1282 ^c^ ± 134	<0.0001
PRINCIPAL COMPRESSIVE STRAIN (µε)					
PH (n = 9)	NA	−1045 ^a^ ± 223	−715 ^b^ ± 223	−1092 ^a^ ± 225	<0.0001
MH (n = 9)	−1275 ^a^ ± 260	−954 ^a^ ± 254	−858 ^a^ ± 260	−1338 ^a^ ± 276	0.068
DH (n = 9)	NA	−1011 ^a^ ± 327	−1272 ^a^ ± 330	−1382 ^a^ ± 346	0.0696
PQ (n = 8)	−1278 ^a^ ± 420	−1627 ^a^ ± 415	−1477 ^a^ ± 424	−1632 ^a^ ± 415	0.0923
MQ (n = 9)	−929 ^a^ ± 202	−899 ^a,b^ ± 204	−1355 ^b,c^ ± 202	−1571 ^c^ ± 204	<0.0001
DQ (n = 9)	NA	−343 ^a^ ± 207	−689 ^b^ ± 218	−692 ^a,b^ ± 229	0.0201
MAXIMUM SHEAR STRAIN (µε)					
PH (n = 9)	NA	1343 ^a^ ± 231	929 ^b^ ± 234	1119 ^a,b^ ± 234	0.0005
MH (n = 9)	1421 ^a^ ± 227	1378 ^a^ ± 246	1297 ^a^ ± 246	1846 ^a^ ± 278	0.2318
DH (n = 8)	NA	2376 ^a^ ± 377	2646 ^a,b^ ± 383	3123 ^b^ ± 415	0.4052
PQ (n = 8)	NA	2196 ^a,b^ ± 560	1865 ^a^ ± 566	2264 ^b^ ± 560	0.0449
MQ (n = 9)	1290 ^a^ ± 241	1354 ^a^ ± 250	1629 ^a^ ± 245	2285 ^b^ ± 256	<0.0001
DQ (n = 8)	NA	1431 ^a^ ± 363	2235 ^b^ ± 366	1935 ^b^ ± 369	<0.0001

**Table 5 animals-13-01872-t005:** Principal Strain Directions Least Squared Means for Load. Least Squared Means Principal Strain directions (°) and standard error for strain directions by load when averaged over all treatments. Within a row, values that do not share a superscript are statistically different at *p* < 0.05. *p* values are Type 3 tests of fixed effects for the load in the ANOVA model. Locations are defined as the proximal heel (PH), middle heel (MH), distal heel (DH), proximal quarter (PQ), middle quarter (MQ), and distal quarter (DQ) (Figure 1B).

Gauge	Load			*p*-Value
	3600 N	4600 N	6200 N	
PRINCIPAL TENSILE DIRECTION (°) lsm ± se				
PH (n = 9)	87 ^a^ ± 12	86 ^a^ ± 12	85 ^a^ ± 12	0.74
MH (n = 9)	82 ^a^ ± 10	84 ^a^ ± 10	79 ^a^ ± 10	0.81
DH (n = 9)	97 ^a^ ± 20	98 ^a^ ± 20	79 ^a^ ± 20	0.08
PQ (n = 8)	83 ^a^ ± 5	82 ^a^ ± 5	81 ^a^ ± 5	0.87
MQ (n = 9)	92 ^a^ ± 13	94 ^a^ ± 13	96 ^a^ ± 13	0.88
DQ (n = 9)	121 ^a^ ± 19	118 ^a^ ± 19	119 ^a^ ± 19	0.96
PRINCIPAL COMPRESSIVE DIRECTION (°) lsm ± se				
PH (n = 9)	177 ^a^ ± 12	176 ^a^ ± 12	175 ^a^ ± 12	0.74
MH (n = 9)	172 ^a^ ± 10	174 ^a^ ± 10	169 ^a^ ± 10	0.81
DH (n = 9)	159 ^a^ ± 17	187 ^a^ ± 17	168 ^a^ ± 17	0.11
PQ (n = 8)	173 ^a^ ± 5	172 ^a^ ± 5	171 ^a^ ± 5	0.92
MQ (n = 9)	182 ^a^ ± 13	184 ^a^ ± 13	186 ^a^ ± 13	0.88
DQ (n = 9)	211 ^a^ ± 19	208 ^a^ ± 19	209 ^a^ ± 19	0.96
PRINCIPAL SHEAR DIRECTION (°) lsm ± se				
PH (n = 9)	132 ^a^ ± 12	131 ^a^ ± 12	130 ^a^ ± 12	0.74
MH (n = 9)	127 ^a^ ± 10	129 ^a^ ± 10	124 ^a^ ± 10	0.81
DH (n = 9)	142 ^a^ ± 20	143 ^a^ ± 20	124 ^a^ ± 20	0.08
PQ (n = 8)	127 ^a^ ± 6	127 ^a^ ± 6	121 ^a^ ± 6	0.37
MQ (n = 9)	137 ^a^ ± 13	139 ^a^ ± 13	141 ^a^ ± 13	0.88
DQ (n = 9)	166 ^a^ ± 19	163 ^a^ ± 19	164 ^a^ ± 19	0.96

**Table 6 animals-13-01872-t006:** Principal Strain Directions Least Squared Means ± Standard Errors. Principal strain directions (least squared means ± standard errors) by nail treatment (no shoe (NS), toe (T), toe plus quarter (TQ), and toe plus quarter plus heel (TQH)), when averaged over all loads. Within a row, values that do not share a superscript are statistically different at *p* < 0.05. *p* values are Type 3 tests of fixed effects for treatment in the ANOVA model. NA = These variables did not have NS included in the model, as the order was found to be significant upon the initial testing of the model. Locations are defined as the proximal heel (PH), middle heel (MH), distal heel (DH), proximal quarter (PQ), middle quarter (MQ), and distal quarter (DQ) (Figure 1B).

Gauge	Treatment				*p*-Value
	NS	T	TQ	TQH	
PRINCIPAL TENSILE DIRECTION (°) lsm ± se					
PH (n = 9)	75 ^a^ ± 12	86 ^b^ ± 12	85 ^b^ ± 12	97 ^c^ ± 13	<0.0001
MH (n = 9)	58 ^a^ ± 10	67 ^a^ ± 11	69 ^a^ ± 11	132 ^b^ ± 13	<0.0001
DH (n = 9)	80 ^a^ ± 20	86 ^a,b^ ± 20	83 ^a,b^ ± 20	115 ^b^ ± 21	0.053
PQ (n = 8)	NA	75 a ± 5	85 ^b^ ± 5	86 ^b^ ± 5	0.002
MQ (n = 9)	56 ^a^ ± 13	101 ^b^ ± 14	105 ^b^ ± 13	114 ^b^ ± 14	<0.0001
DQ (n = 9)	102 ^a^ ± 19	106 ^a^ ± 19	104 ^a^ ± 21	166 ^b^ ± 24	0.031
PRINCIPAL COMPRESSIVE DIRECTION (°) lsm ± se					
PH (n = 9)	165 ^a^ ± 12	176 ^b^ ± 12	175 ^b^ ± 12	187 ^c^ ± 13	<0.0001
MH (n = 9)	148 ^a^ ± 10	157 ^a^ ± 11	159 ^a^ ± 11	222 ^b^ ± 13	<0.0001
DH (n = 9)	173 ^a,b^ ± 18	146 ^a^ ± 17	170 ^a,b^ ± 18	197 ^b^ ± 21	0.045
PQ (n = 8)	NA	165 ^a^ ± 5	175 ^b^ ± 5	176 ^b^ ± 5	0.002
MQ (n = 9)	146 ^a^ ± 13	191 ^b^ ± 14	195 ^b^ ± 13	204 ^b^ ± 14	<0.0001
DQ (n = 9)	192 ^a^ ± 19	196 ^b^ ± 19	194 ^b^ ± 21	256 ^b^ ± 24	0.031
PRINCIPAL SHEAR DIRECTION (°) lsm ± se					
PH (n = 9)	120 ^a^ ± 12	131 ^b^ ± 12	130 ^b^ ± 12	142 ^c^ ± 13	<0.0001
MH (n = 9)	103 ^a^ ± 10	112 ^a^ ± 11	114 ^a^ ± 11	177 ^b^ ± 13	<0.0001
DH (n = 9)	125 ^a^ ± 20	131 ^a^ ± 20	128 ^a^ ± 20	160 ^a^ ± 21	0.053
PQ (n = 8)	NA	120 ^a^ ± 6	125 ^a^ ± 6	131 ^a^ ± 6	0.080
MQ (n = 9)	101 ^a^ ± 13	146 ^b^ ± 14	150 ^b^ ± 13	159 ^b^ ± 14	<0.0001
DQ (n = 9)	147 ^a^ ± 19	151 ^a^ ± 19	149 ^a^ ± 21	211 ^b^ ± 24	0.031

## Data Availability

The data are not in a formal data management store; however, the authors can be contacted for the dataset to be sent as requested.

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
