# Peer review of "Hoof Expansion, Deformation, and Surface Strains Vary with Horseshoe Nail Positions"

_animals, 2023, doi:10.3390/ani13111872_

Round 1

Reviewer 1 Report

This is a good study to be added to the physiology of the equine hoof. The information related to the biomechanics involving the "etiology" of the long toe/low heeled horse is important. Possibly to make this study more clinically appealing adding 1) what percent of shod (race) horses have shoes nailed in the "H" position. 2) expanding from their discussion lines around #337 to add their thoughts on quarter crack etiology.  

Possible typos: line 128 parenthesis. line 404 leave out the first "our".

All the references need reviewing and corrections/consistencies made where needed, e.g., but not limited to 1) Am J Vet Res in #5 & #8; 2) Proc. AAEP in #3 & #18; 3) Floyd, A.E. & Mansmann, R.A. and the book doesn't have 480 pages, 4) are all the page numbers correct in all the references.

Author Response

This is a good study to be added to the physiology of the equine hoof. The information related to the biomechanics involving the "etiology" of the long toe/low heeled horse is important. Possibly to make this study more clinically appealing adding 1) what percent of shod (race) horses have shoes nailed in the "H" position. 2) expanding from their discussion lines around #337 to add their thoughts on quarter crack etiology.  

  • We concur that it would be good to know the percentage of racehorses that are shod with nails close to the heels. Unfortunately, we are unaware of any such data.
  • We have added a sentence in the conclusion to address the potential for quarter crack propagation which now reads, “This expansion of the quarter proximally with the use of nails at the quarter may also be associated with the potential for vertical crack propagation that sometimes occurs in underrun heels.” Also included more generically in the conclusion, “It is possible that abnormalities in hoof wall mechanics could contribute to a variety of hoof wall pathologies, for example, quarter cracks and sheared heels.”

Possible typos: line 128 parenthesis. line 404 leave out the first "our".

Thank you for catching those mistakes. They have been changed in the text as requested.

All the references need reviewing and corrections/consistencies made where needed, e.g., but not limited to 1) Am J Vet Res in #5 & #8; 2) Proc. AAEP in #3 & #18; 3) Floyd, A.E. & Mansmann, R.A. and the book doesn't have 480 pages, 4) are all the page numbers correct in all the references.

1-2) Thank you for pointing out those formatting errors. We have corrected them as suggested and per the journal guidelines.

3) Thank you for catching that error on the page number. It has been corrected now.

4) All page numbers have been checked as suggested for accuracy.

Reviewer 2 Report

Using an array of sensors, the authors comprehensively demonstrate that heel expansion and distal quarter expansion decreased as nails were placed closer to the heel of the hoof.  Specifically, using rosette strain gauges, the authors show that palmar shoe nailing effects an increase in proximal hoof wall tension in an altered direction that could draw the distal heel dorsally with palmar tubules out of alignment with more dorsal tubules. This is a significant finding that should reinforce the already established doctrine that palmar heel nailing is detrimental to foot health.  Why the practice should be discouraged now has a scientific backing. I recommend the paper be published pending attention to the following suggestions.

L41 – is hoof to mean foot in this context – the hoof itself is epidermis without elastic structures.  Separate the usage of hoof when foot is meant.

L41 – unsuitable reference

L44 – the distal hoof wall and the lamellae are primarily responsible for energy absorption and force dissipation within the foot during the stance phase.  Heel expansion has little to do with energy absorption and force dissipation 1

L329 The meaning of this sentence is opaque.  Seems to imply that the laminar junction is confined to the quarters. Re-write aiming for clarity.  Third, greater hoof wall thickness and stabilization of the quarter region as the distal phalanx moves distally and exerts axial traction on the quarters through the laminar junction likely contributes to smaller quarter expansion [13].

333 - The normal circumferential hoof expansion is restricted distally by the nails resulting in the quarters expanding at the proximal aspect since this is the area most able to 340 absorb the diverted force.

L356 These sentences lack clarity.   It is OK say ‘Conformation of the hoof capsule reflects hoof growth at the coronary region’ although it may be more accurate to replace ‘growth’ with ‘production’ since the hoof is adult and has stopped growing.  In what way does ‘Hoof growth both responds to and reflects hoof loading conditions.’ – isn’t hoof growth constant regardless of loading conditions – if there is evidence to the contrary it needs a solid reference.  While hoof conformation is managed at the solar margin, enhancing our understanding of hoof abnormalities requires further investigation of how hoof growth responds to loading and hoof management conditions. What sort of management do you mean? Is it intrinsic and physiological or with nippers and rasp?

L347 – If uniform restriction of expansion and deformations of the heel and distal quarter regions by the additional heel nails explains why TQH nail placement reverses proximal quarter expansion what explains the same thing happening in the unshod foot (where presumably there was no distal restriction. Please clarify.

L353 What is the ‘fulcrum for the coronary band curvature’

L366 – This is an important sentence but is too long and needs full stops. As the hoof wall is fixed at both the coronary band proximally and by the nail in the shoe distally, when the hoof is loaded and the hoof wall is compressed, fixation by the nail distally could be preventing hoof movement and instead cause tension dorsally of the hoof wall as demonstrated by both increased principle tensile strains and a counterclock wise rotation of principle tensile strain directions. Suggest;

When the hoof wall is fixed at both the coronary band proximally and by the nail in the shoe distally, hoof loading compresses the hoof wall.  Our findings of both increased principle tensile strains and a counter clock wise rotation of principle tensile strain directions suggest that fixation of the hoof wall by distal nails prevented hoof movement and caused dorsal hoof wall tension.  

L377. The paragraph ending on L377 is the core of the work.  The results are difficult to describe in words so I suggest a diagram with arrows showing the direction of the pull that results in underrun heels.

L415.  Conclusion should include other factors that contribute to underrun heels – palmar nail placement is not the only factor.  Feet can be trimmed to create long toe and low heel. Many unshod brood mares and feral horses ambulating on soft substrates have underrun heels2.

              1.           Lanovaz JL, Clayton HM, Watson LG. In vitro attenuation of impact shock in equine digits. Equine Vet J Suppl 1998:96-102.

              2.           Hampson BA, Ramsey G, Macintosh AM, et al. Morphometry and abnormalities of the feet of Kaimanawa feral horses in New Zealand. Aust Vet J 2010;88:124-131.

Author Response

Using an array of sensors, the authors comprehensively demonstrate that heel expansion and distal quarter expansion decreased as nails were placed closer to the heel of the hoof.  Specifically, using rosette strain gauges, the authors show that palmar shoe nailing effects an increase in proximal hoof wall tension in an altered direction that could draw the distal heel dorsally with palmar tubules out of alignment with more dorsal tubules. This is a significant finding that should reinforce the already established doctrine that palmar heel nailing is detrimental to foot health.  Why the practice should be discouraged now has a scientific backing. I recommend the paper be published pending attention to the following suggestions.

L41 – is hoof to mean foot in this context – the hoof itself is epidermis without elastic structures.  Separate the usage of hoof when foot is meant.

Thank you for this suggestion and changes have been made to use foot instead of hoof when referring to the structure as a whole.

L41 – unsuitable reference

We have further clarified this reference to include the specific pages that support this information in addition to editing the sentence to include the reference suggested in the next suggestion. “Heel expansion and elastic structures within the hoof foot provide some mecha-nisms for attenuating ground impact forces along with bones and joints in the distal part of the limb [1-2].”

L44 – the distal hoof wall and the lamellae are primarily responsible for energy absorption and force dissipation within the foot during the stance phase.  Heel expansion has little to do with energy absorption and force dissipation 1

Thank you for sharing this reference. We have changed the sentence to better reflect what we were trying to portray and have included this reference where appropriate in the first paragraph, “attaching a rigid shoe to the hoof could impair normal hoof mechanics by limiting hoof expansion during interaction with surfaces.”

L329 The meaning of this sentence is opaque.  Seems to imply that the laminar junction is confined to the quarters. Re-write aiming for clarity.  Third, greater hoof wall thickness and stabilization of the quarter region as the distal phalanx moves distally and exerts axial traction on the quarters through the laminar junction likely contributes to smaller quarter expansion [13].

We appreciate this suggestion and have clarified the sentence to read, “Third, as the distal phalanx moves distally and exerts axial traction on the hoof wall through the laminar junction, the greater contact area of the quarters than the heels with the laminae likely contributes to smaller quarter expansion [13]”

333 - The normal circumferential hoof expansion is restricted distally by the nails resulting in the quarters expanding at the proximal aspect since this is the area most able to 340 absorb the diverted force.

We have removed the word circumferential from the sentence as this does not accurately portray the behavior of the foot as a whole.

L356 These sentences lack clarity.   It is OK say ‘Conformation of the hoof capsule reflects hoof growth at the coronary region’ although it may be more accurate to replace ‘growth’ with ‘production’ since the hoof is adult and has stopped growing.  In what way does ‘Hoof growth both responds to and reflects hoof loading conditions.’ – isn’t hoof growth constant regardless of loading conditions – if there is evidence to the contrary it needs a solid reference.  While hoof conformation is managed at the solar margin, enhancing our understanding of hoof abnormalities requires further investigation of how hoof growth responds to loading and hoof management conditions. What sort of management do you mean? Is it intrinsic and physiological or with nippers and rasp?

The term growth is prevalent in the literature to mean distal growth of hoof wall from the coronary region toward the ground and hence our use of growth in this instance. However, your point is well taken and we have included the phrase (hoof wall production) in parentheses in the text. We have further clarified this section with a new reference  as requested and now reads, “In hooves where there is a medio-lateral imbalance, the side where the hoof bears more load has slower growth compared to the side which experiences less load.[1922] While hoof conformation is managed at the solar margin through trimming and shoeing, enhancing our understanding of hoof abnormalities requires further investigation of how hoof growth responds to loading and hoof management strategies.”

  1. Pleasant, R.S., S.E. O'Grady, and I. McKinlay, Farriery for hoof wall defects: quarter cracks and toe cracks. Vet Clin North Am Equine Pract, 2012. 28(2): p. 393-406.

L347 – If uniform restriction of expansion and deformations of the heel and distal quarter regions by the additional heel nails explains why TQH nail placement reverses proximal quarter expansion what explains the same thing happening in the unshod foot (where presumably there was no distal restriction. Please clarify.

We have clarified our meaning by changing the sentence to now read, “This is likely due to a more uniform restriction of expansion and deformations of the heel and distal quarter regions by the additional heel nails similar to the uniform lack of restriction the hoof experiences when unshod.” 

L353 What is the ‘fulcrum for the coronary band curvature’

This sentence has been removed. This was an editing mistake, and we appreciate you catching the error.

L366 – This is an important sentence but is too long and needs full stops. As the hoof wall is fixed at both the coronary band proximally and by the nail in the shoe distally, when the hoof is loaded and the hoof wall is compressed, fixation by the nail distally could be preventing hoof movement and instead cause tension dorsally of the hoof wall as demonstrated by both increased principle tensile strains and a counterclock wise rotation of principle tensile strain directions. Suggest;

When the hoof wall is fixed at both the coronary band proximally and by the nail in the shoe distally, hoof loading compresses the hoof wall.  Our findings of both increased principle tensile strains and a counter clock wise rotation of principle tensile strain directions suggest that fixation of the hoof wall by distal nails prevented hoof movement and caused dorsal hoof wall tension. 

We have changed the entire paragraph these sentences are in so as to clarify this information more clearly and eliminating this grammatical error. “Several findings associated with the addition of heel nails could contribute to the development of underrun heels. The direction of principal tensile strain in the distal heel and quarter regions changed orientation to exert dorsal traction on the distal aspect of the heel when the hoof is fixed to the ground (Figure 3A). Similarly, the orientation of principal shear strains changed to facilitate shear between the proximal and distal portions of the hoof in a plane perpendicular to the dorsal hoof wall, thus allowing the proximal portion of the heel to move palmarly relative to the distal portion of the heel. Further, the direction of principal compressive strains changes to direct compression on the heel in a distal orientation, potentially forcing the bulb distally. Consistent with this finding is the shortening of the palmar chord on the hoof wall. Additionally, the most extreme limitation of heel expansion occurred at the proximal level of the heel (Figure 3B), which could contribute to additional compression driving the bulb and proximal aspect of the heel distally. These features are consistent with displacement of the proximal aspect of the heel in palmar and distal directions, decrease in heel angle, narrowing of the heel bulbs, and distal displacement of the coronary band in the heel region.”

L377. The paragraph ending on L377 is the core of the work.  The results are difficult to describe in words so I suggest a diagram with arrows showing the direction of the pull that results in underrun heels.

A new figure has been added to the discussion to show the direction of the pull in tensile forces as well as the direction change for shear forces and expansion variables that shorten under the TQH condition that together can lead to the development of an underrun heel.

L415.  Conclusion should include other factors that contribute to underrun heels – palmar nail placement is not the only factor.  Feet can be trimmed to create long toe and low heel. Many unshod brood mares and feral horses ambulating on soft substrates have underrun heels2.

 We have added an additional sentence to include these potential factors that reads, “While nail placement may have a role in the development of an underrun heel, other factors may also contribute to this development including trimming and shoeing methods, surface substrate, genetics, and exercise.”

  1. Lanovaz JL, Clayton HM, Watson LG. In vitro attenuation of impact shock in equine digits. Equine Vet J Suppl 1998:96-102.
  2. Hampson BA, Ramsey G, Macintosh AM, et al. Morphometry and abnormalities of the feet of Kaimanawa feral horses in New Zealand. Aust Vet J 2010;88:124-131.

Reviewer 3 Report

The reviewer appreciates the work the authors have contributed to in the area of equine research. This study is novel and original, and has merit to being published after some minor revisions.

Author Response

The reviewer appreciates the work the authors have contributed to in the area of equine research. This study is novel and original, and has merit to being published after some minor revisions.

Simple summary: Very good, only suggest one modification. At the end of the paragraph, it is stated that “changes in compression and tension may cause abnormal changes in hoof shape”. If the authors have the room, would also like to see a mention of horse soundness/performance/functionality.

Thank you for this suggestion. We have modified the last sentence to now read, “Over time the observed changes in compression and tension may cause abnormal changes in hoof shape that can affect soundness and performance.”

Abstract:

Line 25 – Remove “the” before underrun, may consider combining the first two sentences.

This has been changed as suggested.

Line 30 – Modify to “in vitro limb loading of cadaveric limbs”.

This has been changed as suggested.

Lines 34-37 – Include p-values corresponding to results.

The P-value has been added where appropriate and now reads, “Nails placed palmar to the quarters of the hoof decreased heel expansion (P<0.001).”

Introduction:

Line 42 – Remove extra space after “forces.”.

This has been changed as suggested.

Line 47 – Modify to “during racehorse training”.

This has been changed as suggested.

Line 58 – Remove extra space after “wall.”.

This has been changed as suggested.

Line 64 – 70 – The authors may consider placing the objectives first, followed by the hypothesis.

This has been changed as suggested.

Materials and methods:

Line 77 – “walk to canter loads” while it has been otherwise specified in the document that tests were performed at walk, trot, and canter loads, the current wording sounds like loads are applied to mimic a transition from walk to canter.

Thank you for pointing this out. The sentence now reads, “The effects of nail placement for horseshoe attachment on heel and quarter expansions, hoof wall (global) deformations, hoof wall (local) surface strains and fetlock extension were assessed during loading of cadaveric equine limbs in vitro to simulate the middle of stance for walk, trot, and to canter loads.”

Line 80 – Remove extra space after “outcomes.”.

This has been changed as suggested.

Line 87 – 89 –The reviewer understands that obtaining cadaver forelimbs may be difficult, and requests an explanation of the various breeds in the discussion, and potential breed effects.

A sentence has been added to the limitations paragraph that now reads, “Hoof dimensions including toe and heel angles, hoof wall thickness and overall size vary among breeds. Breed was not considered in the current study and could potentially influence hoof strains and deformations.”

Figure 1 – A good graphic that contributes to the understanding of the methods.

Thank you.

Line 112 – Remove extra period at end of paragraph.

This has been changed as suggested.

Line 151 – Is there a source for the canter peak vertical force? The current reference is placed in a way that it does not appear to refer to the canter peak vertical force.

Thank you for catching the misplaced reference. It has been changed to the end of the sentence.

Line 159 – Did the one limb that did not reach 6700 N fail at or after 6200 N?

No it did not fail. While we had set the MTS to load up to 6700N, data was not captured past 6200N for the one limb, so to allow for use of the limb in the study and to have all of the data to be analyzed for the same loads, the 6200N load was used as all limbs achieved this load.

Line 174 – Please include SAS version.

This has been added to the manuscript.

Line 179 – Was the effect of breed considered? (stock-type vs. non-stock type). Was the effect of leg (R or L) considered?

Breed was not considered as our focus was on riding horses in general and keeping a consistent hoof size to accommodate the same size shoe. Effects of L or R were considered and were not significant and thus not discussed.

Results:

Tables – The authors may consider an SEM column/row when appropriate, to make differences between means easier to identify. The authors should align the largest means with the first superscript (a), and proceed further down the alphabet as means decrease.

We have adjusted the tables to help them to read better.

Table 1, 3, & 5 – The authors may consider aligning the load headings with gaits they represent.2

While we appreciate this suggestion, 6200N does not represent a true canter load and is the reason we chose to list these tables with the corresponding load rather than with the gait.

Lines 217 – 218 – This current wording sounds like a transition was evaluated. In table 1, gaits are not currently referred to, only loads. The text and the table should align by both referring to loads or to gaits.

We are unsure of what is being referred to in this suggestion. Both the tables and the sentence at these lines refer to load but not to gaits. We made the change earlier as suggested to clarify that the data were collected at specific load points that coincide with specific gaits, but were not captured as part of a transition of gait.

Line 244 – Remove extra space after “strains.”.

This has been changed as suggested.

Discussion:

Line 347 – Remove space after “condition.”.

We are unable to find this space. We have gone through the manuscript to fix spaces that were added in as a result of using the template provided by the journal.

Line 371 – is the counterclockwise direction regardless of a right or left limb?

No, which is why we specify the orientation on a right limb for discussion. The direction change of the strains is from a distalpalmar to proximodorso direction. We used the terminology of counterclockwise when oriented on a right hoof, lateral aspect when considering the diverse readership of the journal and to simplify the explanation of the direction of strain change.

Reviewer 4 Report

Thank you for the interesting manuscript. I have listed my concerns/questions by line number:

87-89: Why not try for the same breed and age of horses with similar sized feet to reduce variability? Were the feet roughly the same shape prior to trimming and shoe placement? Was the coronary band in a single plane in all of the feet?

158: What about the 4600 N (trot) load?

329-331: Did you consider selecting feet with a similar hoof wall thickness?

393-395: While the mid stance is the greatest load, the heel landing would probably provide the most information on heel expansion. How could you address that?

Author Response

Thank you for the interesting manuscript. I have listed my concerns/questions by line number:

87-89: Why not try for the same breed and age of horses with similar sized feet to reduce variability? Were the feet roughly the same shape prior to trimming and shoe placement? Was the coronary band in a single plane in all of the feet?

While we concur that uniformity in breed and age would reduce unwanted variability in the study outcomes, we were limited to cadavers that had permission for unrestricted use and were free from known lameness issues. Within this population of cadavers, variability was reduced by standardizing selection of cadaver limbs that had a hoof size and shape that fit the same size manufactured horseshoe. While no hoof is exactly the same shape as another hoof (they vary in shape even within the same horse), having all hooves of a size that allowed for a proper fit of the same size shoe (fitted to the widest part of the frog so that the weightbearing hoof wall was covered by the shoe) allowed us to keep similarity in hoof size among the different breeds.

 Conformation of the hooves were within normal limits and without abnormalities before trimming and shoe placement. The coronary band was level (did not bend at an angle) in all hooves.

158: What about the 4600 N (trot) load?

Thank you for catching that error. It has been changed to reflect the actual data points that were used for the analysis.

329-331: Did you consider selecting feet with a similar hoof wall thickness?

We did not measure hoof wall thickness. The hooves were all the same size as they all fit a size 7 Queen XT shoe that is a mass-produced shoe used in the racing industry. This ensures that the nail holes are over the white line and the shoe covers the weight bearing portion of the hoof wall. Horses that did not fit this criterion were not included in the study.

393-395: While the mid stance is the greatest load, the heel landing would probably provide the most information on heel expansion. How could you address that?

For this study we wanted to investigate the point of greatest hoof expansion which is during midstance. Heel strike and the time to midstance does provide different information on how the heels behave as the hoof is loaded and would best be captured in a more dynamic setup where the limb is allowed to land and rotate such that it mimics the behavior of the foot during heel strike through toe off.

Reviewer 5 Report

Regardless of whether this study can lead to advances in the preventive of veterinary treatment of horses, it provides an interesting assessment of the impact of nail placement on force distribution in shod hoofs. The extensive tables provide a concise but thorough overview of all pertinent results. I encourage the authors to present their main results, the differential force distribution in different treatment and force groups, in a graphical way, and thus increase the potential impact of their work. It would be good to visualize the main variation in a bar plot.

The descriptions are mostly precise, and the general layout of the manuscript is easy to follow. Some of the methods need to be explained in greater detail (see comments below), especially the order or treatment, and size range of hoofs are important factors in evaluating the soundness of this study. Furthermore, I encourage the authors to switch to a more active sentence structure (e.g. ‘previous studies found’).

Overall, this study is intriguing, and pending additional clarifications of methods will provide a solid base for further investigations into hoof injury.

43: Start a new sentence with “However,”

48: For the non-veterinarian, briefly define the terms “underrun” and “hoof conformation”.

50: Make this a presence active sentence: “A study on thoroughbred horses showed that odds of carpal effusion increase as the heel becomes more underrun.”

64: Presence: “We hypothesize …”

79: Please explain in greater detail. Was every treatment assessed for every specimen? Was the order of treatments randomized, or universal? My review is based on the assumption that all hoofs were tested in the same order: NS-T-TQ-TQH-NS; please confirm this in the text or provide an alternate order.

88: What size are the hoofs? The fit of the horseshoe to the hoof, and the distance of the nail to the edge of the hoof likely impacts force distribution. Reporting the range of width and length measurements of the hoofs will suffice.

98: instrumented: do you mean “measured”? Why was the hoof shod, then nails removed, then measured, then shod again? Using preexisting holes for nails may impact force distribution.

170: Strain and stress were measured directly. Why was anything interpolated? Are some of the results based on linear models, instead of empirical data?

188: Please indicate in your result tables how many specimens were included in each analysis (n per row). You clearly laid out the reasons and mechanisms for excluding specimens, but we don’t know how many observations were actually excluded in this way.

Fig. 1B) How is the direction of 0 degrees defined?

244, and throughout the Results: please directly reference the associated table.

307: Directly reference the associated table.

381: Would that technique be as precise as yours for measuring strains?

393: You applied three different loads, associating them directly with different gaits. That is in itself problematic, partially because different gaits not only impose different peak loads but also loading patterns onto the hoof. The direction of the ground reaction force differs markedly between a canter and a gallop. Your fixed direction of force induction actually circumvents problems associated with gait selection, which is an advantage you can stress.

You can be a little more optimistic in your description of potential shortcomings, and advertise approaches like the ones mentioned in line 381 for future studies.

Considering previous studies of mobility in ungulate feet, how much of an impact would you expect three-dimensional rotation to have on stress distribution? I expect that the limitations associated with your 2D study are relatively small.

The Tables look great, and provide a thorough overview of the results. The distribution of values, in particular, is valuable, and is too often ignored in veterinary studies. If the authors could provide a graph with the most interesting results (e.g. force differences between treatments), this would further enhance the accessibility of these results.

Author Response

Regardless of whether this study can lead to advances in the preventive of veterinary treatment of horses, it provides an interesting assessment of the impact of nail placement on force distribution in shod hoofs. The extensive tables provide a concise but thorough overview of all pertinent results. I encourage the authors to present their main results, the differential force distribution in different treatment and force groups, in a graphical way, and thus increase the potential impact of their work. It would be good to visualize the main variation in a bar plot.

While we concur with the desire to present information graphically, we struggled with how to present the quantity of information graphically in an easily interpretable way – other than the strain gauge findings.

The descriptions are mostly precise, and the general layout of the manuscript is easy to follow. Some of the methods need to be explained in greater detail (see comments below), especially the order or treatment, and size range of hoofs are important factors in evaluating the soundness of this study. Furthermore, I encourage the authors to switch to a more active sentence structure (e.g. ‘previous studies found’).

Overall, this study is intriguing, and pending additional clarifications of methods will provide a solid base for further investigations into hoof injury.

43: Start a new sentence with “However,”

This change has been made as suggested.

48: For the non-veterinarian, briefly define the terms “underrun” and “hoof conformation”.

A new sentence has been added after the first sentence of this paragraph that reads, “One type of abnormal hoof conformation is the underrun heel, defined as a heel angle at least 5° lower than the toe angle.”

50: Make this a presence active sentence: “A study on thoroughbred horses showed that odds of carpal effusion increase as the heel becomes more underrun.”

The sentence has been changed as suggested.

64: Presence: “We hypothesize …”

The sentence has been changed as suggested.

79: Please explain in greater detail. Was every treatment assessed for every specimen? Was the order of treatments randomized, or universal? My review is based on the assumption that all hoofs were tested in the same order: NS-T-TQ-TQH-NS; please confirm this in the text or provide an alternate order.

To clarify how the nail treatments were ordered, we have now added, “The order of the 3 nail treatments was different among individual limbs so that variance associated with the effect of order of treatment (e.g., potential stress relaxation of the cadaveric limb during repeated testing) could be partitioned out from the variance associated with the effect of nail placement to minimize the effect of test order on statistical nail treatment outcomes.”

88: What size are the hoofs? The fit of the horseshoe to the hoof, and the distance of the nail to the edge of the hoof likely impacts force distribution. Reporting the range of width and length measurements of the hoofs will suffice.

The width and length of the hooves were not measured. The hooves were of similar size because hooves were selected because they accommodated a size 7 Queen XT shoe that is a mass-produced shoe used in the racing industry. This ensures that the nail holes are over the white line and the shoe covers the weight bearing portion of the hoof wall. Horses/limbs that did not fit this criterion were not included in the study. Using a standard pre-manufactured shoe minimizes the variance of hoof size.

98: instrumented: do you mean “measured”? Why was the hoof shod, then nails removed, then measured, then shod again? Using preexisting holes for nails may impact force distribution.

Instrumented was correct as the hoof markers and strain gauges were placed on the hooves to obtain measurements. To clarify, the sentence has been modified to read, “Then the shoe and nails were removed, and the limb and hoof were instrumented with markers and gauges.”

The shoe was removed as an unshod hoof needed to be tested before and after the nail treatments to determine if stretch relaxation occurred in the limb during the study. Under the section Study Design in the methods, we describe the purpose of this method, “In addition, NS treatments were included before and after the assigned nail treatments for comparison with the nail treatments and to determine if repeated limb loading affected outcome measures (ideally the outcome measures for the first and last NS treatments would not be statistically different).”

A sentence has been added to clarify the use of holes that were used and now reads, “Due to the location of the hoof markers and strain gauges, the need to maintain the shoe in the same position on the hoof for all treatments, and the likelihood that driving nails in an instrumented hoof would detach the strain gauges from the hoof, the same nail holes were used from when the shoe was initially fitted for all nail treatments.”

170: Strain and stress were measured directly. Why was anything interpolated? Are some of the results based on linear models, instead of empirical data?

Strain and marker displacement data were acquired at 20 Hz and 60 Hz while each limb was loaded to 6700N at 0.25 Hz (~3350N/s) (with exception of one limb that was only loaded to 6200N). Consequently, strain and marker displacement data were not necessarily captured at exactly 1800, 3600, 4600 or 6200N load levels. Therefore, a forecasting technique was used in Excel to determine the strain and marker displacement data that corresponded as closely as possible to 1800, 3600, 4600 or 6200N loads by linear interpolation between data captured at the nearest load points on either side of the specific load level.

188: Please indicate in your result tables how many specimens were included in each analysis (n per row). You clearly laid out the reasons and mechanisms for excluding specimens, but we don’t know how many observations were actually excluded in this way.

Thank you for pointing out this missing information. It has been added to each table as requested.

Fig. 1B) How is the direction of 0 degrees defined?

Clarification has been made to the figure to describe this better and now reads, “The 90/270° axis is along the tubules such that the 0/180° axis is perpendicular to the tubules.”

244, and throughout the Results: please directly reference the associated table.

The table was referenced at the end of the paragraph as the whole paragraph refers to that table. We have moved the reference to the table to after the first sentence in that paragraph to refer the reader to the table earlier. References to the associated tables throughout the results are placed at the end of the first sentence referring to the table. If the editor wishes for us to refer to the tables at the end of each sentence, we are glad to make that change.

307: Directly reference the associated table.

This line is within the discussion and typically references to tables are not placed in the discussion section. If the editor would like for us to reference the tables further in the discussion section, we would be happy to make that change.

381: Would that technique be as precise as yours for measuring strains?

A sentence has been added to address this and now reads, “DIC provides similar strain results though perhaps less precise due to the nature of the inconsistency of the application of the speckle pattern, however, this method would show the distribution of strains over the entire speckled surface of the hoof wall and could be useful for further studies.”

393: You applied three different loads, associating them directly with different gaits. That is in itself problematic, partially because different gaits not only impose different peak loads but also loading patterns onto the hoof. The direction of the ground reaction force differs markedly between a canter and a gallop. Your fixed direction of force induction actually circumvents problems associated with gait selection, which is an advantage you can stress.

Thank you for this suggestion. We have added an additional sentence to reflect this information which now reads, “Additionally, with the fixed direction of force during the simulation of mid-stance of gait, variation that occurs with the different ground reaction force patterns associated with each gait is not a factor in the study.”

You can be a little more optimistic in your description of potential shortcomings, and advertise approaches like the ones mentioned in line 381 for future studies.

Thank you for this suggestion.  “DIC provides similar strain results though perhaps less precise due to the nature of the inconsistency of the application of the speckle pattern, however, the DIC method would also show a larger perspective of strains on the surface of the hoof wall as a whole and could be useful for further studies.” We hope with the addition of this information, a more positive tone has been achieved.

Considering previous studies of mobility in ungulate feet, how much of an impact would you expect three-dimensional rotation to have on stress distribution? I expect that the limitations associated with your 2D study are relatively small.

 Fundamentally, our findings are limited to the middle of stance when vertical ground reaction forces are expected to be maximal. The direction and magnitude of ground reaction forces and thus stress distribution on the hoof change throughout stance and can be different for each gait. While the study provides an important window into the effects of nail placement on hoof deformations, the study is limited to the mid-stance condition.

The Tables look great, and provide a thorough overview of the results. The distribution of values, in particular, is valuable, and is too often ignored in veterinary studies. If the authors could provide a graph with the most interesting results (e.g. force differences between treatments), this would further enhance the accessibility of these results.

We appreciate the advantage of graphically depicting data results. However, we struggled how to graphically depict the large amount of data in this study. Figure 2 provides this information with differences between treatments with regard to strain magnitude and direction changes in a polar plot format with the plots aligned with each strain gauge placement. An additional figure has been added to show the overall picture of the changes with the nails in the most palmar location (Figure 3).

Round 2

Reviewer 3 Report

The authors have made the requested modifications to the manuscript which have improved the presentation of the methods and results greatly. This manuscript now appears ready for publication.

Author Response

Thank you so much for taking the time to review our work. It is greatly appreciated.

Reviewer 4 Report

Thank you for responding to my concerns and questions.

Author Response

(The authors gave the same response as above.)

Reviewer 5 Report

The authors have addressed all criticism previously raised, and I fully support publication, after a few additional changes are made. Please see below for two minor problems and one request for additional reference of results.

Table 1: this is a copy of Table 2. Please replace it with the actual Table 1.

Table 6: please reformat this table.

369: Directly reference the corresponding tables in this sentence, as it provides a great overview over your findings.

Author Response

The authors have addressed all criticism previously raised, and I fully support publication, after a few additional changes are made. Please see below for two minor problems and one request for additional reference of results.

Table 1: this is a copy of Table 2. Please replace it with the actual Table 1.

Thank you for catching that mistake. We have changed the table to be the appropriate table referenced.

Table 6: please reformat this table.

Thank you for catching that formatting problem. There seems to be some problem with the template that changes the tables when pasted into the document. We have re-inserted the table and it appears to have been corrected.

369: Directly reference the corresponding tables in this sentence, as it provides a great overview over your findings.

We are unsure what line you are referring to. Line 369 is a blank space between Tables 5 and 6.